# Degradation and Stabilization of Resin-Dentine Interfaces in Polymeric Dental Adhesives: An Updated Review

Faiza Amin [1], Muhammad Amber Fareed [2], Muhammad Sohail Zafar [3,4], Zohaib Khurshid [5,*], Paulo J. Palma [6,7,*] and Naresh Kumar [8]

1 Science of Dental Materials Department, Dow Dental College, Dow University of Health Sciences, Karachi 74200, Pakistan; faiza.ameen@duhs.edu.pk
2 Department of Adult Restorative Dentistry, College of Dentistry, Gulf Medical University, Ajman 4184, United Arab Emirates; prof.mafareed@gmu.ac.ae
3 Department of Restorative Dentistry, College of Dentistry, Taibah University, Al Madina, Al Munawwarra 41311, Saudi Arabia; mzafar@taibahu.edu.sa
4 Department of Dental Materials, Islamic International Dental College, Riphah International University, Islamabad 44000, Pakistan
5 Department of Prosthodontics and Dental Implantology, College of Dentistry, King Faisal University, Al-Ahsa 31982, Saudi Arabia
6 Center for Innovation and Research in Oral Sciences (CIROS), Faculty of Medicine, University of Coimbra, 3000-075 Coimbra, Portugal
7 Institute of Endodontics, Faculty of Medicine, University of Coimbra, 3000-075 Coimbra, Portugal
8 Science of Dental Materials Department, Dr. Ishrat Ul Ebad Khan Institute of Oral Health Sciences, Dow University of Health Sciences, Karachi 74200, Pakistan; kumar.naresh@duhs.edu.pk
* Correspondence: zsultan@kfu.edu.sa (Z.K.); ppalma@uc.pt (P.J.P.); Tel.: +966-558-420410 (Z.K.); +351-239-249-151 (P.J.P.)

**Abstract:** Instability of the dentine-resin interface is owed to the partial/incomplete penetration of the resin adhesives in the collagen fibrils. However, interfacial hydrolysis of the resin-matrix hybrid layer complex activates the collagenolytic and esterase enzymes that cause the degradation of the hybrid layer. Adequate hybridization is often prevented due to the water trapped between the interfibrillar spaces of the collagen network. Cyclic fatigue rupture and denaturation of the exposed collagen fibrils have been observed on repeated application of masticatory forces. To prevent interfacial microstructure, various approaches have been explored. Techniques that stabilize the resin–dentine bond have utilized endogenous proteases inhibitors, cross linking agents' incorporation in the exposed collagen fibrils, an adhesive system free of water, and methods to increase the monomer penetration into the adhesives interface. Therefore, it is important to discover and analyze the causes of interfacial degradation and discover methods to stabilize the hybrid layer to execute new technique and materials. To achieve a predictable and durable adhesive resin, restoration is a solution to the many clinical problems arising due to microleakage, loss of integrity of the restoration, secondary caries, and postoperative sensitivity. To enhance the longevity of the resin-dentine bond strength, several experimental strategies have been carried out to improve the resistance to enzymatic degradation by inhibiting intrinsic collagenolytic activity. In addition, biomimetic remineralization research has advanced considerably to contemporary approaches of both intrafibrillar and extrafibrillar remineralization of dental hard tissues. Thus, in the presence of biomimetic analog complete remineralization of collagen, fibers are identified.

**Keywords:** biomimetic; bond degradation; dental adhesive; dentine-resin interface; resin composites

## 1. Introduction

Restorative and adhesive dentistry has witnessed extraordinary improvements after the innovations in contemporary adhesive materials. These new adhesive systems do not require any mechanical retention through features such as dovetails, grooves, sharp

internal angles, and undercuts [1]. For the success of modern restorative dentistry, these adhesive systems play a critical role, as sound tooth structure would be preserved using these newer systems. In addition, by using these contemporary and advanced adhesive systems, secondary caries due to microleakage may be reduced or eliminated [1].

Buonocore, in 1955, reported that enamel and dentine surfaces could be made more receptive to adhesion by altering it through acid pretreatment. Moreover, he revealed that by conditioning the human enamel with 85% phosphoric acid, the acrylic resin could be bonded with enamel. Simultaneously, this technique was used for pit and fissure sealants and class III and class V restorations. Until the 1950s, developments in dentine adhesives were sluggish. '*Surface active comonomer*', synthesized by Bowen, demonstrated that resins formed a chemical bond to dentinal calcium, but the commercial products developed on the basis of this co-monomer resulted in very poor clinical performance [2]. In the dentinal bonding systems over the past 45 years, a lot of variations have been discovered in terms of the chemistry of the bonding agent, the effectiveness of the bonding agent, the technique, and the mechanism. Due to the continuously increased demand of esthetic bonded restorations, an increase in the evolution of bonding agents occurred accordingly [3]. The paradigm shift of esthetic dentistry has transformed adhesive dentistry in the past few years and has gained much consideration due to extensive research on dental adhesives and continuously changing concepts. Prompt progress in adhesive dentistry is due to the increased demand for minimally invasive tooth restorations and esthetics [4]. Different types of dental adhesives have emerged due to this research and development. These adhesive systems are prone to degradation of the resins. Therefore, the longer-term success of clinical stability and durability of resin-based polymers depends upon many factors, primarily on the degradation of the dentinal collagen. Although there have still been some uncertain hitches in the adhesive system in the past 50–60 years, it is incredible to witness adhesive bonding. The performance of the adhesive polymers cannot be comprehended accurately, as they contain a complex mixture of various constituents. For the correct clinical use of these adhesives, sound and intense knowledge is required. The literature shows variable clinical data in which low bonding strength was observed for some materials but higher bond strength in other materials [5]. Bond failure of these resin polymers over time might be due to the elution of unreacted monomers, water sorption, and polymer swelling [6].

Thus, this review consulted several studies and the factors important for the degradation of adhesive resins and the strategies to eliminate these factors. The aim of this review was to present the factors that are responsible for the resin-bond interface degradation, and the possible strategies to minimize and prevent this degradation. In addition, the role of biomimetic remineralization and associated factors in preventing the degradation of the resin adhesive interface was explored.

## 2. Composition of Dental Adhesives

By composition, dental adhesives consist of resin monomer solutions [7]. These monomers consist of both hydrophobic and hydrophilic groups. The interaction and copolymerization of the adhesive system with the restorative material are by the hydrophobic component, whereas the hydrophilic component improves the wettability of the material with the hard dental tissues [8]. Other components in the adhesive system are comprised of initiators, inhibitors, stabilizers, organic solvents, and inorganic fillers [8]. However, to achieve durable bonding with these adhesive techniques, the composition and structure of enamel and dentine need to be examined. Besides water and organic material, enamel is composed of crystalline hydroxyapatite, having a high-energy surface with strong intermolecular forces. On the contrary, dentine is a biological composite of hydroxyapatite, having a collagen network with low surface energy and intermolecular forces. Structurally, dentine is entirely different from enamel, and it is humid and less stiff than enamel. Dentine contains a smear layer, organic contents, and dentinal tubules [9]. In contrast to the enamel, dentine undergoes an aging physiological process, due to which dentine permeability

decreases and the thickness of dentine increases [10]. The bonding technique for enamel is different from dentine bonding due to the fact that the enamel becomes dried easily. The loss of dentine bond strength is due to the degradation of hydrophilic resin constituents and due to the degradation of collagen fibrils. This degradation will damage the hybrid layer. To avoid collagenolysis in the resin-dentine interface, extensive research activity was conducted to gain further understanding of the role of enzymes in the hybrid layer. This review emphasized various factors accountable for the degradation of collagen fibrils and the hybrid layer, in addition to it discussing the strategies to prevent and control the hydrolytic enzyme-related loss of the bond strength of the adhesives and the damage of the hybrid layer (HL).

## 3. Classifications of Dental Adhesives

Due to the complex nature of adhesive agents, the concept of generations has been used by the dental industry and academia. Dental adhesive systems have evolved from no-etch to total-etch to Self-Etch (4th, 5th, 6th, 7th, and 8th generation) techniques, and the details of these are presented in Tables 1 and 2. While improving the chemistry of the adhesive systems, each subsequent generation has focused on minimizing the number of steps to simplify and reduce the procedural time, and they achieved a faster application technique, which is required during clinical applications.

**Table 1.** A description of the components of bonding agents.

| Components | Ingredients | References |
|---|---|---|
| Resin components | HEMA, Bis GMA, TEGDMA, is the main component of adhesives systems. Different monomers, cross-linkers and functional polymer group, methacrylamides and MDPB, methacryloxylethylcetyl ammonium chloride, PENTA | [11–13] |
| Functional monomer | 4 META and 10 MDP and GPDM | [12] |
| Photo-initiators | Camphorquinone, 1-phenyl-1,2 propanedione (PPD), MAPO and BAPO | [14,15] |
| Chemical initiators | Benzoyl peroxide and tertiary amine | [16,17] |
| Inhibitors | Butylated hydroxytoluene and others | [18,19] |
| Solvents | The most common solvents are water, acetone, Solvent effect adhesion via wettability, collagen expansion and monomer ionization | [20,21] |
| Fillers | Bioactive fillers, nanofillers, fluoride releasing fillers Fillers to improve radio-opacity, montmorillonite nano-clay | [22,23] |
| Recent modification | Glutaraldehyde as the denaturation of collagen in dentine and the occlusion of the dentinal tubules MMPs, chlorhexidine, bioactive ingredients, and antimicrobial and remineralizing agents | [24–26] |

HEMA: 2-Hydroxyethyl methacrylate, BIS-GMA: Bisphenol A-glycidyl-methacrylate, TEGDMA: Triethylene glycol dimethacrylate, MDBP: 12-Methacryloloxydodecyl pyridinium bromide, PENTA: Dipentaerythritol penta-acrylate phosphate, 4 META: 4-Methacryloxyethyl trimellitate anhydride, MDP:10-Methacryloloxydecyl dihydrogen phosphate, GDPM: Glycerol-Phosphate dimethacrylate, MAPO: (4-maleimidophenyl)oxirane, BAPO: (2,4,6 trimethyl benzoyl)-phosphine oxide, MMPs: Matrix Metalloproteinases.

**Table 2.** Summary of the different classification systems used for categories of dental adhesives.

| Classification | Description and Characteristics of Several Adhesive Systems | References |
|---|---|---|
| Current adhesive system | **Etch and Rinse (total etch)**, Etchant removed the smear layer to form demineralized dentine<br>Complete infiltration of monomers is not achieved leaving incompletely infiltrated zones. | [27–30] |
| | **Self Etch**, Separate acid-etch step was not needed<br>Stability is dependent on the coupling between the collagen fibril substrate and the adhesive resin<br>Reduced porosities, homogenous resin infiltration, and better collagen fibrils protection<br>Immunohistochemical labeling with anti-type I collagen antibodies presents a weak, uniform hybrid layer<br>The efficacy of bonding to enamel without the need for separate acid etching is questionable | [30–34] |
| Number of steps | **Three step (4th Generation)**, Involves etch, prime, and bond (three bottles)<br>Highest in bond strength and greatest durability | [35] |
| | **Two step (5th generation)**,<br>Etch, prime, and bond in a single coating (two bottles)<br>Simplified method<br>Efficient and stable bonding less predictable and more difficult dentine bonding | [35] |
| The historical concept of dental adhesive generations<br>Historical concept of dental adhesive generations<br>*(continued)* | **First generation adhesives (One-step)**<br>Contains glycerophosphoric acid dimethacrylate (NPG-GMA)<br>Ionic bond with hydroxyapatite and covalent bond to collagen<br>Smear layer not removed, polymerization shrinkage occurs, low bond strength (2–6 MPa) | [36–38] |
| | **Second generation adhesives (One-step)**,<br>Polymerizable phosphates incorporated to bis-GMA resins to enhance bonding<br>Formation of ionic bond in calcium and chlorophosphate groups<br>The smear layer was not removed<br>Debonding, microleakage, and low bond strength 4–6 MPa are disadvantages | [2,37–39] |
| | **Third generation adhesives (One-step)**,<br>Acid-etch enamel and dentine to partially eliminate the smear layer<br>A primer application after the acid rinsed away with water<br>Greater bond strength then first and second | [39–42] |
| | **Fourth generation adhesives (Three-step)**<br>Golden standard in dentine bonding<br>Complete removal of smear layer<br>Total-etch technique and concept of hybridization introduced were introduced<br>Technique sensitive due to complexity of multiple bottles and steps, and it was time-consuming<br>Bond strength is in 10–20 MPa range and reduced margin leakage | [39,43–47] |

**Table 2.** *Cont.*

| Classification | Description and Characteristics of Several Adhesive Systems | References |
|---|---|---|
| | **Fifth generation adhesives** (Two-step), Etch and Rinse concept, combining the primer and adhesive resin into one application<br>Hybrid layer formation, more prone to water degradation than 4th generation adhesive<br>Bond strength 3–25 MPa | [45,48,49] |
| | **Sixth generation adhesives** (Two Steps), Self-Etching primers<br>Does not involve a separate etching step<br>Efficacy is less dependent on the dentine hydration than the total-etch systems<br>Sufficient bond strength to conditioned dentin while the bonding with enamel was less effective | [39,45,50,51] |
| | **Seventh generation (One-step)**, One-bottle Self-Etching system<br>Prone to hydrolysis, waster sorption, and chemical breakdown<br>Limited resin infiltration and creates some porosities<br>Lowest initial and long-term bond strengths | [49,52–55] |
| | **Eight Generation (One Step)**, Nano-filler (12 nm) present Self Etch generations<br>An acidic hydrophilic monomer<br>Improved enamel and dentin bond strength and stress absorption and produced a longer shelf life<br>Increased resin monomer penetration and improved hybrid layer thickness<br>Due to nano-fillers clusters, it can form cracks and reduce the bond strength | [56–59] |

Dental adhesives are generally characterized in historical generations that reflect the amended handling performance for improvements in novel preparations rather than new adhesion concepts or mechanisms. There are two major adhesive concepts based on chemistry and the mechanism of adhesion to the tooth structure:

a.  Superficial demineralization of dentine and enamel, which depend on the complete removal of the smear layer.
b.  Partial or superficial dissolution of the smear layer in the adhesive interface to create a hybrid layer [27].

Both models encourage resin adhesion by micro-mechanical retention to enamel and dentine. However, a supplementary chemical bond to the substrate is also present in both concepts [28]. Van Meerbeek et al. [29] proposed an adhesive classification into two categories, Etch and Rinse, and the selective etch technique. A distinct acid-etching step was not required in Self Etch systems since the adhesive resins simultaneously infiltrate and demineralize the tooth structure and create a more homogenous infiltration of adhesive resins in the demineralized collagen fibrils [30]. The bond strength of the Self Etch adhesive technique, therefore, depends on the collagen fibril and the resin adhesives coupling [31,32]. However, the effectiveness of Self Etch bonding to enamel without an acid-etch step is still questionable [30,31].

**4. Degradation of Adhesive Interface/Hybrid Layer**

The adhesion of direct restoratives to the tooth structures is due to the ongoing development and recent research in adhesive systems. In spite of the developments in chemistry, composition, and classification, the stability and strength of the resin-dentine interfacial bond remain questionable [60]. The consequences of resin degradation at the adhesive interface results in post-operative sensitivity, marginal staining, and secondary caries, due to which the durability of the restoration will be compromised [40]. This might

be due to the fact that degradation will produce marginal deterioration and weaken the adhesion procedure [61]. It is a commonly established fact that dentine-resin adhesive interface degradation occurs after the use of current dental adhesives and has been an interesting research topic most recently. The following factors are proposed to influence the degradation of the hybrid layer and resin-dentine interface.

### 4.1. Degradation of Adhesive Resins

The main factor involved in the chronic degradation of the adhesive resins is the hydrolysis and adhesive resin leaching from the resin-dentine matrix [62,63]. Water diffusion into the hydrophilic adhesive initiates the leaching process, and, due to the adhesive phase separation, there is a limited degree of polymerization in the hydrophilic domains [64]. In the aqueous environment, the poorly polymerized hydrophilic phase undergoes degradation more quickly. The primary factor involved in the reduction of the bond strength of the resin adhesive interface is the hydrolysis within the hybrid layer, which contributes to poor adhesion after some time [64]. Water begins to penetrate the resin restoration interface after prolonged exposure of the resin restorations to oral fluids. The water acts as a plasticizer between the polymer chains of the adhesives and as a molecular lubricant. This molecular lubricant will cause mechanical wear of the exposed adhesives [64]. This allows for greater transport of both water and enzymes, along with increasing the surface area and abrading the resin dentine interface surface, leading to the acceleration of matrix degradation [38].

The other factor that enhances the degradation of the resin interface is chemical hydrolysis due to the water transport or salivary fluids in methacrylate materials, which results in damage to the ester bonds [38]. An immediate increase in the bond strength was observed after the infiltration of the exposed collagen fibrils by the hydrophilic 2-hydroxyethyl methacrylate (HEMA) monomer [63,65]. The limitation of these adhesive systems is that they compromise the longevity of the dentine-resin bond [66]. One of the most important factors besides the presence of water, that contributes to the degradation of adhesive resins, is the incomplete polymerization of variable degrees that can be associated to the extent of fluid movement in the adhesive hybrid layer [5,62,63,67]. Contemporary dental adhesive systems comprise both hydrophilic and hydrophobic components. Hydrophobic monomers continue to stay on the surface, whereas the hydrophilic components infiltrate the interior of the hybrid layer [60]. These systems produce heterogeneous resin layers due to the nanophase separation ration phenomenon [68].

One of the most significant causes involved in the degradation of the resins is the hydrophobic camphorquinone (CQ) photo-initiator [69]. The hydrophobic CQ initiator may potentially cause a suboptimal degree of conversion of hydrophilic monomers [69], resulting in deficient polymerization in the hybrid layer deep zone [69]. In that instance, to improve the degree of conversion of adhesives, it is suggested to use camphorquinone in addition to water-compatible photo-initiators, such as TPO (ethyl 4-dimethylaminobenzoate and diphenyl (2,4,6-trimethylbenzoyl)-phosphine oxide) [69]. This photo-initiator possibly reduces the damaging effect of the nanophase separation by increasing the degree of conversion of the hydrophilic and the hydrophobic components of the resin polymer [68,70,71]. Other minor causes of resin interface degradation include the expansions and the contraction in resin restorative materials due to temperature changes and occlusal forces. These factors compromised the dentine resin bond stability by allowing the penetration of oral fluids and water into the resin interface [72]. Moreover, hydrophobic and hydrophilic cytotoxic by-products, such as ethylene glycol and methacrylic acid, are released because of the breakage of ester bonds present in the HEMA [73]. This ester bond breakage also occurs when saliva, pulp, and bacteria release esterase enzymes [74].

### 4.2. Degradation of Collagen

Dayan et al. [75] and Tjäderhane et al. [76] have reported the collagenolytic activity in dentine. In aseptic conditions, collagen can degrade over time, as confirmed by Pashley et al. [77], because of the intrinsic matrix proteases. It was reported that treated specimens

with enzyme inhibitors remarkably found a decrease in the intrinsic dentine gelatinolytic and collagenolytic activity [77]. Thenceforth, scientists put their efforts towards exploring the role of these enzymatic activities in the degradation of the HL, enzymes involved in the degradation, and their localization within the dentine, and they also investigated the approaches to reduce or stop this enzymatic activity [77]. The matrix metalloproteinase and cysteine cathepsins are the most noticeable collections of endogenous enzymes within dentine, and they are discussed below.

4.2.1. Matrix Metalloproteinase (MMP)

MMPs are proenzymes common to both bone and dentine. They are $Zn^{2+}$ and $Ca^{2+}$-dependent endogenous proteases that consist of a bridge between the $Zn^{2+}$ ions and the cysteine residue [78]. The tertiary structure of MMPs is preserved by their $Ca^{2+}$ part, whereas the $Zn^{2+}$ ions are responsible for the enzyme activation. In the intact form, these MMPs prevent the binding of the $Zn^{2+}$ ions with the water molecules, thereby preventing enzyme activation (Figure 1) [78]. Their classification is based on the substrate on which they act similar to collagenases (MMP-1, -8, -13, and MMP-18), gelatinases (MMP-2 and MMP-9), stromelysins (MMP-3 and MMP-10), matrilysins (MMP-7 and MMP-26), and membrane-type MMPs (MMP-14, -15, -16, and MMP-24). MMPs that are present in human dentine are MMP-2, -3, -8, -9, and MMP-20 [79].

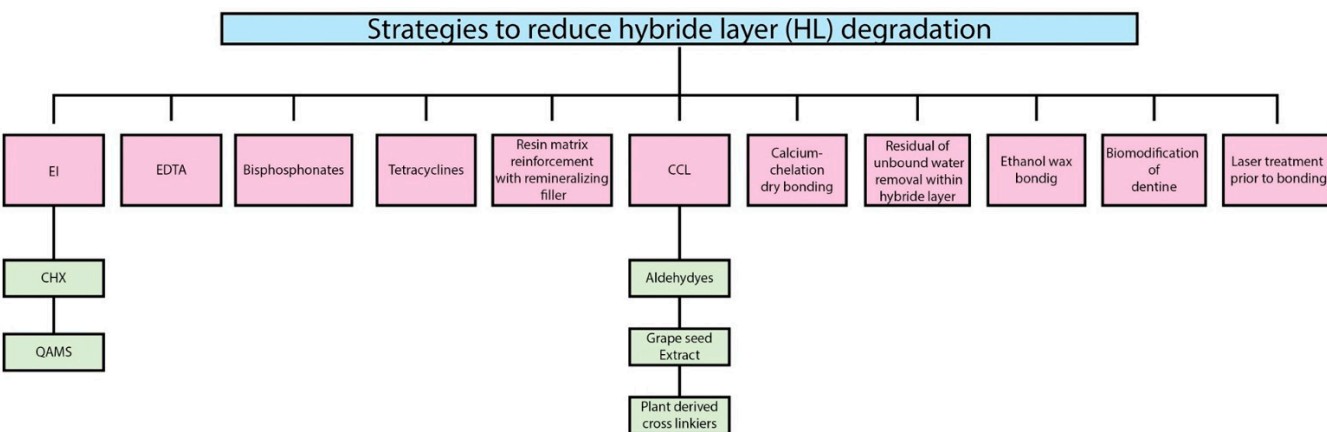

**Figure 1.** Summary of the strategies used to reduce hybrid layer degradation. EI → Enzymes Inhibitors; CHX → Chlorhexidine; QAMS → Quaternary ammonium methacrylate compounds; EDTA → Ethylene diamine tetra acetic acid (EDTA); CCL → Collagen Cross Linkers.

During dentine maturation, MMPs play an important role, but they become inactive after the collagen matrix mineralization is completed [80]. In addition, during growth, organogenesis, and normal tissue turnover, MMPs are responsible for the degradation of extracellular matrix proteins [81]. Nevertheless, the comparative influence of such enzymatic activity on bond degradation is still indistinguishable. The most relevant pathway of bond degradation is thought to be due to hydrolytic water sorption [81]. Although there is inconsistent literature available on the different kinds of adhesives about enzyme exposure and activation, the data regarding retarding/arresting bond degradation effects upon use of MMP inhibitors vary among studies [81]. The most plentiful MMPs found in dentine are MMP-2 and MMP-9 [81]. Mazzoni et al. [82] studied MMP-2 and MMP-9 molecular forms in the demineralized dentine by immunoassay, gelatin zymography, and western blotting [82] and reported that the organic matrix of the fibrillar network of human dentine consists of both MMPs as intrinsic constituents [82]. It was also found that both mineralized and un-demineralized dentine have distinctive distribution patterns and concentrations of MMPs, and they concluded that the bioavailability and activation of MMPs are affected by the demineralization [61]. The roles of MMPs in the Etch and Rinse technique and the Self Etch approach were compared in a study after mixing adhesives with

human dentine powder, and it was reported that, when such adhesives were applied to acid-etched dentine, there was an increase in the quantity of activated, non-denatured enzymes and proteolytic activities in the tested adhesives [83,84]. However, the denaturation of MMPs can occur during the conditioning of dentine at a lower pH (0.7–1) of phosphoric acid [85]. Latent forms of the enzymes activated by low pH trigger the cysteine switch and employ their effect on the catalytic action of dentine. The inhibitory activity of tissue inhibitors of MMP also decreases through this process. However, other studies showed the disparity and demonstrated that, after mixing demineralized dentine powder with Self Etch adhesives, the low pH value was neutralized rapidly in the Self Etch approach, resulting in momentarily preventing the MMPs activity [86]. Many in vivo and in vitro techniques can be used to activate the MMPs. In vitro techniques can be physical (heat and low pH) and chemical (chaotropic agents, sulfhydryl groups, and reactive oxygen). In vivo activation can be by proteases and other MMPs, and they are associated with MMP activation, which was induced by the dentinal adhesive application [87]. Lehmann et al. [88] found that, in human odontoblasts, the synthesis of MMP-2 was increased after the application of the adhesive, due to which movement through the hybrid layer to dentinal tubules also increased [89].

### 4.2.2. Cysteine Cathepsins

In dentine matrices, the prominent endogenous proteolytic enzymes that are involved in the degradation of dentine are cysteine cathepsins (CTs). The presence of CTs in dentine was reported to compromise 98% of the cathepsin activity against dentine collagen [90,91]. These proteases are expressed by mature human odontoblasts cells and pulpal tissues. In human dentine, there are 11 CTs that have been found involving the CT-K, CT-L, and CT-B [92]. They can generate multiple collagen fragments due to the presence of glycosaminoglycan (GAG) side chains, as they are able to cleave helical collagen at multiple sites, whereas other CTs and MMPs can only cleave the non-helical telopeptide part of the collagen cleave [93]. The association of cysteine cathepsins with caries progression and hybrid layer degradation was demonstrated by a few researchers [90,91,94]. MMPs and cysteine cathepsins are located near the target substrate and are close to each other. Therefore, there is synergistic activity between them that was found in sound and carious dentine. In this way, the two different classes of these proteases worked as a cascade network [90,91,94]. Dentine-bounded MMPs were further activated by the acidic activation of cysteine cathepsins [95]. In addition, the spectrofluorometric use of fluorogenic substrates was carried out for these proteases (MMP and CT activities), and it was found that, in matrix degradation, two types of proteases have diverse coordinated roles. Interestingly, in dentine CT-MMP interactions, it was found and demonstrated that MMPs and CTs regulate the activities of each other [91].

### 4.3. Incomplete Infiltration of the Resin Adhesives

Incomplete hybridization of the adhesive in the collagen complex in the Etch and Rinse technique is a result of the difference between penetration of the adhesive and action of the conditioning acidic agents [96]. Nanopercolation occurs because of the incomplete hybridization of collagen networks, as they become more susceptible to hydrolytic degradation [96]. A stable and complete hybrid layer cannot be achieved such as in the inter- and intrafibrillar compartments; the monomers are not able to replace the free and collagen-bound water [97,98]. In addition, large monomers such as BIS-GMA were entrapped in inter-fibrillary spaces due to highly hydrated proteoglycan hydrogels. These proteoglycan hydrogels only allow small monomers such as HEMA to penetrate toward the base of the hybrid layer [99]. As HEMA creates weak linear chains, they will cause cyclic fatigue failure of the collagen chains when subjected to stress [100].

## 5. Strategies to Reduce Hybrid Layer (HL) Degradation

Several strategies have been reported in the literature to improve the longevity and decrease the degradation of the hybrid layer by using protease inhibitors, enzyme inhibitors, and collagen cross-linkers as primary agents [101–103]. Some of these agents that influence the endogenous dentinal enzymes are discussed in the section below.

### 5.1. Enzymes Inhibitors

5.1.1. Chlorhexidine (CHX)

Chlorhexidine is the most extensively investigated enzymatic inhibitor and an antimicrobial agent and is characterized by the excellent inhibition of MMP activity in both dentine and resin [104]. Due to chelating properties at low concentrations (0.2%), it showed excellent inhibition of protease enzymes [104–109]. The literature regarding the mechanism of action of CHX is scarce, but it might be because of the cationic behavior of CHX, as it binds to both mineralized and unmineralized dentine [105–108]. As CHX consists of chlorine, it prevents its hydrolytic activity by binding to the zinc of the catalytic domain of MMPs [104]. When CHX digluconate (0.5 wt.%, 2.0 wt.% and 4.0 wt.%) was incorporated in experimental etch-and-rinse adhesives, no adverse effects on the degree of conversion were found [110]. However, Cadenaro et al. [111] demonstrated that, although the degree of conversion was not affected by the incorporation of 1% or 5% CHX into the adhesive resins, the elastic modulus was significantly decreased. Furthermore, bond strength was preserved for 12 months without affecting the ultimate tensile strength, solubility, DC, and water sorption after the incorporation of CHX in Etch and Rinse adhesives [112]. Da Silva et al. [113] incorporated Galardin, Batimastat, GM1489, and CHX as enzyme inhibitors in Etch and Rinse adhesives and found similar results in CHX and GM1489.

5.1.2. Quaternary Ammonium Methacrylates Compounds (QAMS)

The mechanism of action of quaternary ammonium compounds is similar to CHX, as both are positively charged. The most prominent and already tested MMP inhibitor in the quaternary ammonium compounds is benzalkonium chloride (BAC). BAC consists of various alkyl chains and is a combination of alkylbenzyl-dimethylammonium chlorides. These compounds showed favorable bond strength results over time in comparison to CHX, as they strongly bind to demineralized dentine [114–116]. To evaluate the mechanical properties of the unfilled resins after incorporating quaternary ammonium methacrylates, Hoshika et al. [117] found that the degree of conversion of these resins was improved, but wet toughness and ultimate tensile strength were decreased by the addition of 10% QAMs, whereas when 15% QAMs were added, it decreased the mechanical properties of the resins significantly [117]. Therefore, it was documented that, although the degree of cure will be increased, the ultimate tensile strength and Young's modulus were decreased in QAM containing adhesive [117]. 12-meth-acryloyl-oxydodecyl-pyridinium bromide (MDPB) is a quaternary ammonium methacrylate QAM and has been incorporated into several contemporary adhesive systems. It is well-known for its antimicrobial properties in the adhesive resins, MDBP polymerize with methacrylate; therefore, leaching of this compound was restricted and can serve as a microbe [118]. It has an excellent capability of preventing MMP activity [118–121]. Further research is needed to demonstrate that adhesives based on QAM compounds increase the strength of the resin-dentine interface by effectively inhibiting MMPs at the resin dentine interface and also to focus on the development of quaternary ammonium functionalities [122].

### 5.2. Ethylene Diamine Tetra Acetic Acid (EDTA)

For decades, due to the chelating properties, ethylene-diamine tetra acetic acid (EDTA) has been used in endodontics. EDTA binds to $Zn^{2+}$ ions from the catalytic site of the MMPs and removes the $Ca^{2+}$ from the collagen matrices [123,124]. However, a long application time and the reversibility caused by water solubility are the main drawbacks [125].

### 5.3. Bisphosphonates

Bisphosphonates are proteases inhibitors that act by chelating zinc and calcium ions from several enzymes [126]. In particular, good instant outcomes were observed with polyvinylphosphonic acid (PVPA), but with uncertain durability [127]. Tezvergil-Mulutuay et al. [127] used polyvinylphosphonic acid (PVPA) on recombinant MMP-9 and found that it efficiently inhibited this gene with less hydroxyproline release. The bonding between PVPA and collagen is electrostatics. PVPA can become trapped in collagen through 1-ethyl-3-(3-dimethyl aminopropyl) carbodiimide (EDC) [128]. This phenomenon makes PVPA more advantageous in terms of bond durability when compared with CHX. All these outcomes suggested that, to enhance the durability and longevity of resin dentine bonds, PVPA can be incorporated. However, there is a dearth of literature about bisphosphonates as MMP inhibitors; thus, future work should be performed.

### 5.4. Tetracycline

Tetracyclines along with their analogues doxycycline and minocycline are broad-spectrum antibiotics used as effective MMP inhibitors, having cationic chelating properties. [129,130]. Doxycycline decreases dentine matrix degradation intensely [131]. Chemically-modified tetracyclines (CMT-3, Metastat) are exceptionally efficient in reducing MMPs in dentine caries and can retain their MMP-inhibiting capacity, although the antimicrobial activity is not ideal [132]. They act on enzymes by inhibiting their activity and secretions. In addition, they are also involved in calcium chelation [133]. However, improvements in the dentine-resin bond by the tetracycline have not yet been evaluated. These compounds need further investigation due to their potent MMP inhibiting potential. Though, during photo-oxidation, these compounds can induce the purple stain of teeth and are therefore not considered suitable for clinical use.

### 5.5. Collagen Cross Linkers

In dentine, collagen matrix cross-linking is not only a natural mechanism used to increase the mechanical properties of dentine, but it also is used to make dentine less susceptible to enzymatic and hydrolytic degradation [134,135]. Several studies, therefore, have reported the incorporation of numerous chemical constituents that have cross-linking properties in dental adhesives [134,135]. Cross-linking agents stabilize the collagen structure and make it more resilient to enzymatic degradation by attaching to the amino-carbonyl groups of the collagen [134,135]. These cross-linkers prevent the hybrid layer degradation through several mechanisms, as mentioned below:

a. Firstly, cross linkers disable the degeneration process by changing the molecular mobility of the MMPs [134,135].
b. Secondly, they render this collagen less prone to hydrolysis by creating supplementary cross-links between the molecules of collagen [134,135].
c. Lastly, many different mechanisms have been involved in the inhibition of MMPs and CTs by these cross-linkers.

These mechanisms could be evading the cysteine switch oxidation, which in the substrate cleavage sites will be hidden by these cross-linkers, and the dysregulation of endogenous protease expression, within collagen protection of cleavage sites, by cross-linkers and the inactivation/silencing of proteases activity [136].

### 5.5.1. Aldehydes

Aldehyde (glutaraldehyde) is widely reported in the literature as a strong cross-linking agent for its use in dentistry [135,137–140]. However, it is rarely used in clinical dentistry because of its cytotoxicity [139,141]. It forms a covalent bond between the amino groups of peptidyl lysine and hydroxylysine residues within the collagen [139,141]. To improve and enhance the mechanical durability of dentine, the simplest unsaturated aldehyde known as Acrolein (acrolein (2-propenal)) was used as an additional primer strength preservation

of dentine resin interface [142]. As with the glutaraldehyde, Acrolein is also proved to be cytotoxic and inadequate for clinical practice.

5.5.2. Grape Seed Extracts

The agents with low cytotoxicity have been widely investigated, such as crosslinkers derived from natural grape seeds known as proanthocyanidin [143–145] and carbodi-imides [103,146]. In contrast to the aldehydes, they can be easily used in daily clinical practice due to the fact that they are biocompatible and are not cytotoxic [134]. In dentistry, 1-ethyl-3-(3-dimethylamino-propyl) carbodiimide (EDC) is the most extensively studied carbodiimide [143–145]. Ionized carboxyl groups in proteins react with the functional group of EDC and form an intermediate compound known as O-acylisourea. This intermediate product forms a steady covalent-amide bond between the two proteins by reacting with lysine and hydroxylysine amino groups to aid amino cross-linkages with collagen. EDC has a dual function; firstly, in collagen, EDC cross-links with both telopeptide and helical domains, whereas, at the same time, EDC prevents telopeptidase activity [147]. A similar bond strength was found in the resin dentine interface preservation for both EDC and GD [148], but in contrast, EDC showed much lower cytotoxicity. The stiffness of both the demineralized dentine matrix as well as the hybrid layer has been well documented with EDC [149–151]. It was demonstrated that, when EDC was applied on acid etch dentine and stored in artificial saliva for 1 year, it deactivated MMPs efficiently. Dentine powder was studied in this research with the help of zymographic assays [148], and three-dimensional in situ zymographic images were obtained by confocal microscopy [103,146]. However, the use of EDC is limited to clinical practice as it needs a relatively long time (1 h) to cross-link to collagen [152]. Similarly, the cytotoxic effects of 5% GA and EDC in varying concentrations on odontoblast-like cells with dentine barriers were studied by Scheffel et al. [153], and they found that lower concentrations of EDC (0.1, 0.3, and 0.5 M) and 5% GA did not produce trans-dentinal cytotoxic effects on odontoblast-like cells.

5.5.3. Plant-Derived Cross-Linking Agents

The most widely reported plant-derived cross-linking agents used in dentistry are genipin, tannins (polyphenolic compounds), oligomeric proanthocyanidins, and curcumin due to their high potency and low cytotoxicity [102,139,145,154]. These cross-linking agents reinforced the hybrid layer and improved the resin-dentine bonds by reacting with degrada-tion products, resulting in the late polymerization of adhesive resin [155]. These antioxidant substances that possibly prevent MMP activity in the dentine also promote non-enzymatic collagen cross-linking. The elastic strength increases with an increase in the degree of cross-linking of the collagen [156]. Dentine biomodification increases the mechanical properties of demineralized dentine through nonenzymatic collagen cross-linking [157,158]. Proan-thocyanidins are derived from grape seeds and improve the tenacity and elastic modulus of demineralized dentine [159]. However, they are considered unsuitable clinically due to their lengthier (10–60 min) application duration [138]. Moreover, brown pigmentation in dentine, as well as polymerization inhibition of resin monomers due to a decrease in the de-gree of conversion, has been observed with these agents [160,161]. The cross-linking effect is also accomplished by photochemical treatment and physical agents apart from chemical compounds [160,161]. These treatments and agents could be drying, heating [162], and ultraviolet A (UVA) [163] and gamma irradiation. Glutaraldehyde [116], EDC [146,147,164], and similar cross-linking agents [165,166] derived from plants have the capabilities to react with collagen-degrading enzymes.

*5.6. Residual or Unbound Water Removal within the Hybrid Layer*

One of the greatest challenges of highly cross-linked resin polymers is that they undergo phase changes due to their poor solubility in water [64]. Therefore, manufacturers prepare commercial adhesive systems in a variety of solvents such as ethanol to warrant the single phase of resin adhesives during clinical application. When adhesives containing

solvent are applied on moist acid-etched dentine, microscopic phase changes are observed in the adhesives. Pashley et al. [167] demonstrated the solution to this problem by replacing the water rinse with ethanol in the wet-bonding technique, which resulted in dentine saturation by ethanol, not with water. Similarly, Tay et al. [168] reported that, when bis-GMA was applied to ethanol-saturated dentine, excellent resin–dentine bonding was achieved. The danger of phase separation can be avoided completely by using ethanol containing adhesives to dentine treated with ethanol by reducing the residual water at the dentine-resin interface [169]. Due to water absence in the hybrid layer, the collagen matrix may not be cleaved by matrix proteases. However, in the water-wet bonding approach, hydrophobic infiltration of resin adhesives is much less than in ethanol wet-bonding [20,170]. It is well documented that demineralized dentine and ethanol can replace and remove unbound water [171]. Jee et al. [98] determined in their study whether bound water can be replaced by collagen matrices and found that the tightly bound first layer and the second layer of water cannot be replaced by ethanol in collagen, although most of the bound water in the outermost layer was substituted by ethanol.

### 5.7. Calcium-Chelation Dry Bonding

The acid etchant (37% phosphoric acid) used during the adhesive process completely demineralized the collagen fibrils because of its molecular weight, which is around 100 Da, which can easily pervade throughout collagen fibrils [172,173]. Molecules with a molecular weight smaller than 600 Da can enter collagen fibrils easily, and all molecules with a molecular weight larger than 40 KDa cannot permeate into collagen fibrils easily [169,170]. Keeping this point in mind, studies have used calcium chelator (sodium polyacrylate) instead of phosphoric acid [172,173]. They used 15 wt.% calcium chelators with a molecular weight of about 225,000 Da, which is large enough to permeate collagen. After 30 s of chelation, water rinsing and air-drying were carried out, due to which the reaction was stopped. This will lead to the preservation of inter-fibrillar gaps for monomer diffusion inward and those absorbed into the hybrid layer, and only apatite mineral was removed from the extrafibrillar space [174]. In this approach, collagen fibrils remained too stiff to shrink or collapse even after the removal of residual water, as the collagen fibrils remained completely mineralized [174].

### 5.8. Biomodification of Dentine

To enhance the physical and mechanical properties of dental hard tissue, advancements in biomodification have been made by modifying the biochemistry of these dental hard tissues by incorporating or inducing physical agents [134]. One of the most significant physical agents used as biomodification is photo-oxidative techniques [134]. This technique utilized ultraviolet light, which requires the most reactive and unstable type of oxygen singlet for activation. This type of oxygen singlet can be provided by vitamin B2 (riboflavin). Cross-linking occurs between the proline and hydroxyproline of side chains (carbonyl groups) and the glycine of a collagen chain (amino group). These oxygen singlets form covalent bonds when activated by ultraviolet light [134].

### 5.9. Ethanol Wet Bonding

The prime factor involved in the adhesive bond strength durability is the hydrolytic degradation of adhesives. Hydrophilic and ionic monomers have been added to these adhesives to ensure the proper hybridization of wet collagen matrix [8,175,176]. Mechanical properties of these hydrophilic adhesives were found to be compromised, as these polymers contain ester linkages and they are susceptible to water sorption and/or hydrolysis [8,175,176]. A 12-month in vivo study conducted by Brackett et al. [177] concluded that, despite preserving the adhesives with CHX, water-related loss of nano-fillers was observed. In a few studies, loss of bond strength was found despite containing CHX and other enzyme inhibitors [8,175,176]. This loss of strength in adhesive resins might be due to polymerized hydrophilic adhesives, water sorption, or adhesive monomer degradation.

These studies concluded that if water is eliminated from the bonded interface, then water hydrolysis of peptide bonds in collagen and ester-bonds in adhesive polymers might possibly be reduced. This has been the main objective of introducing the concept of ethanol-wet bonding [177]. The phenomenon behind this concept is that acid-etched demineralized dentine matrices that were dehydrated by the ethanol aided the penetration of higher hydrophobic monomers into the interfacial dentine and reduced the collagen hydrophilicity [177]. Ethanol-wet bonding wheedles the infiltration of hydrophobic monomers to demineralize collagen with restricted matrix shrinkage [167]. Water sorption/solubility and resin plasticization decrease because of the infiltration of hydrophobic monomers. In addition, it has been suggested that improved durability of the resin bond occurs, resulting in decreased enzyme-catalyzed hydrolytic collagen degradation because of the elimination of residual water [35,178], and the hybrid layer generated with ethanol-wet bonding also resulted in outstanding durability of resin bond strength and an almost complete absence of nano-leakage [179].

*5.10. Resin Matrix Reinforcement with Remineralizing Fillers*

As in the previous section, it was discussed that not only enzymatic degradation will degrade the hybrid layer, but chemical degradation also played a significant role in the degradation of adhesive durability. Fillers and nanoparticles are included in the top priority list to use as reinforcing adhesives materials [180]. Several studies have demonstrated an increased bond strength and enhanced mechanical and physical properties of the adhesives after the incorporation of copper [180], carboxylic acid-functionalized titanium dioxide [181], silver micro-fillers [182], and zinc oxide [183] nanoparticles. To improve the bond strength of the commercial three-step etch-and-rinse adhesive system (Scotchbond™ 3M ESPE, St Paul, MN, USA), Zirconia nanoparticles were incorporated by Lohbauer et al. [184] into the primer or adhesive. The incorporation of these particles resulted in improved resistance to the hydrolytic process, which might increase the durability of the dentine-resin bond. The rate of the bond degradation will reduce when hydrolysis is diminished due to the reduced water uptake that retarded the proteases activity, leading to the formation of a stronger hybrid layer [161]. Nanotubes are a hexagonal network of carbon atoms that are extremely strong and stiff and have excellent thermal and electrical properties, and they were also incorporated as fillers to resin-based restorative materials to reinforce the resin matrix [185] and thus the resin-dentine bond strength [186,187]. The incorporation up to 20 wt.% of nanotubes in the etch-and-rinse adhesive system and up to 10 wt.% in Self Etch adhesives system have resulted in increased bond strength [183]. The most fascinating quality of nanotubes is the possibility of expanding the cylindrical hollow structure as a medium for the encapsulation of therapeutic molecules as well as protease inhibitors [188]. Feitosa et al. [188] reported that an inhibitor of MMPs (doxycycline) was encapsulated into nanotubes and then incorporated into an adhesive resin, and it was able to inhibit MMP-1 activity without compromising bond strength results. Similarly, nanotubes can be used as a vehicle for the encapsulation of biomimetic agents to prevent bond degradation due to the release of MMP inhibitors, antioxidants, and collagen crosslinkers [188]. Although dental adhesive reinforcement with nanoparticles is an excellent strategy, the beneficial effects due to nanoparticle incorporation are very vulnerable due to agglomeration or the inhomogeneous dispersion of nanofillers in the resin phase, which may reduce the bond strength and physical stability of the adhesive materials [188]. Usually, there is a threshold for filler loading into adhesives, and it depends on the composition of the adhesives and filler type [189].

*5.11. Laser Treatment Prior to Bonding*

For bond strength enhancement, laser irradiation of enamel/dentine has been utilized [190,191]. Erbium-doped Yttrium Aluminum Garnet (Er,YAG) [190,191] and plasma-based lasers [192,193] are the two key sources of lasers that are employed in dentistry. The key factors that contribute to the success of dental bonding are the smear layer removal,

organic content, and water evaporation, and an increased surface area was obtained by Er,YAG laser irradiation [191]. Preceding bonding when dentine is treated with non-thermal atmospheric pressure plasma (NTAPP) laser, it was found that it aided with improved immediate bond strength as well as long-term bond stability after ageing [192]. This might be because carboxyl and carbonyl groups are grafted by the laser application onto the dentinal substrate, which will enhance the chemical and mechanical interaction of the resin monomers. Moreover, positive effects were found when oxidizing agents were used on the dentine before laser application [190], and the authors found a greater resin-dentine bond strength when the Er,YAG laser application on dentine was performed after bleaching. It was demonstrated that increased releases of free radicals were found after the laser during bleaching and made the surface of dentine such that it received adhesives monomers in a better tactic. This will increase the durability of the dentine and resin bond strength. Correspondingly, an efficient method to enhance the resin dentine bond strength is utilizing a non-thermal argon plasma laser for 30 s on sodium hypochlorite-treated dentine [193]. This increase in strength might be due to increased hydrogen bonding interaction between collagen fibrils and adhesive resins after the dentine etching [193].

### 6. Biomimetic Remineralization

In recent decades, biomimetic has developed as a multi-disciplinary approach in dentistry. There are several biomimetic approaches utilized in the field of restorative dentistry; an example is a tooth that was restored using bioinspired peptides, bioactive biomaterials, and biomimetic tissue regeneration to achieve remineralization [194]. To improve the strength properties of adhesive materials, developments in the contemporary adhesive materials and understanding at the nanoscale of biomaterial–tissue interaction are continuously investigated and evaluated [194]. The most ideal, novel, and exciting approach to prevent the collagen fibrils from degradation is biomimetic remineralization. Biomimetic remineralization involves the leaching of ion-releasing materials that simulate the natural remineralization process [195,196]. This process removes the residual water from water-rich regions and intrafibrillar spaces of the hybrid layer by inactivating proteases and reducing collagen degradation. These phenomena increase and restore the strength of the hybrid layer by replacing fibrils with apatite crystallites as well as preserve the durability of the resin-dentine bond interface by preventing the exposed collagen from external challenges [195,196]. There are two types of biomimetic remineralization that occur in adhesive dentistry:

(a) The first approach cannot occur in demineralized dentine where apatite crystals are absent. In this type of remineralization, the remaining mineral crystals act as templates for the regrowth of apatite crystals [197].

(b) The second type of biomimetic remineralization involves incorporating polyanions (polyacrylic acid/polyaspartic acid) and apatite nucleation, resulting in biomimetic remineralization [197].

The carious lesions may lead to the exposure of the collagen fibrils due to the loss of minerals from the dentine. This will lead to the degradation of the collagen fibrils and the deterioration of the mechanical properties of the dentine [196]. Moreover, during restorative and adhesive procedures, various type of techniques and agents are used, such as acid etching, acidic monomers, and chelating agents, that cause the demineralization of dentine and enamel. In addition, partial infiltration of collagen fibrils with resin monomers will cause the micro-permeability and nano-leakage of hybrid layers, especially in the contemporary adhesive system [197]. In addition these systems are not able to remove collagen fibrils, further compromising the properties of the polymeric adhesive system. Due to the incomplete or partial infiltration of resin monomers, these hybrid layers consist of numerous water-filled regions [197]. Insoluble collagen fibrils are slowly solubilized by these that are water-filled. Remineralizing reagents such as nanometer-sized apatite crystallites can be incorporated into these water-filled voids. Polyanions act as templates for specific calcium-binding to promote the nucleation of appetites [197]. Moreover, intrafibril-

lar and interfibrillar remineralization of dentin collagen fibrils were very well demonstrated after the application of non-collagenous protein (NCP) analogues [197].

### 6.1. Amorphous Calcium Phosphate (ACP)

When collagen fibrils were completely demineralized, amorphous calcium phosphate (ACP) was used to achieve the biomimetic remineralization of adhesive resins [198]. To induce growth and nucleation of appetite, this ACP will bind with collagen and serve as a template. In order to produce crystals of apatite with the length that is the best fit in the collagen gap, clusters of polyanions are incorporated around the ACP [198]. This phenomenon is a self-limiting process that takes place inside the water trees (water-filled voids) in the hybrid layer. ACP will slowly and gradually release the apatite in these water trees to mineralize the dentine and enamel. Nevertheless, clinically, this process does not yet seem to be useful, as in vitro lateral diffusion mechanisms were used for the incorporation of the crystals in demineralized dentine [29]. From the practical point of view, this in vitro concept must be translated into clinical trials; although it still has a long way to go, once it is clinically acceptable, these open doors will pave a new revolution in adhesive dentistry. The ossification of MMPs and the remineralization of collagen fibrils occur when hydroxyapatite permeates into these water-filled spaces. Tay and Pashley [199] used ACP in Portland cement for the remineralization of the collagen network by the deposition of apatite crystals and found a meta-stable amorphous calcium-phosphate in these cements. Numerous studies utilized such kinds of biomimetic analogs in the adhesive resins [97,200,201], and they found that the extrafibrillar mineral deposits were formed when mineralization was achieved without the biomimetic analogues. Similarly, biomimetic analogues were directly bonded to the collagen network instead of biomimetic phosphoproteins into the solution and dentinal collagen complete remineralization in a few months [202,203].

### 6.2. Phosphoproteins Analogues

Phosphoproteins analogues play a very significant role in the maturation and mineralization and process for biomimetic remineralization. Without these analogues, incomplete maturation and mineralization occurs [97,200,201,204]. When adhesives (Etch and Rinse [202]), Self-Etch adhesives [203], and primers were incapacitated with phosphoprotein biomimetic analogue investigators observed after 3 months of storage, Etch and Rinse adhesives were found to be increased in microtensile bond strength, and after 6 months of storage, Self Etch adhesive increased in bond strength. Because of many limitations, this technique is not a routine practice in dentistry. When the biomimetic approach was applied to Etch and Rinse adhesives, both intrafibrillar and extrafibrillar compartments were occupied by the apatite crystals, whereas when this approach was applied in Self Etch adhesives, only intrafibrillar spaces were deposited by the apatite crystals [205,206]. For the remineralization of completely demineralized dentine, usually 3–4 months are required.

### 6.3. Polyvinylphosphoric Acid Analogue

Better mechanical properties of the dentine can be obtained using polyvinyl phosphonic acid as biomimetic analogue. This strategy inhibits the endogenous MMPs of dentine through biomimetic remineralization and prevents collagen degradation [128]. It was a documented fact that the strength of the resin dentine interface decreases after 12 months of storage prior to biomimetic remineralization, but the strength after the remineralization long term storage will prevent the durability of the adhesive resin bond [207]. Although these treatments need to be optimized and take additional time for application, they also promise a durable bond over time due to the inactivation of the endogenous proteases of dentine [60].

### 6.4. Bioactive Silicates

One of the strategies to induce biomimetic remineralization is to combine hydrophilic, biodegradable polymers with sodium/calcium phosphosilicate bioglass (bioactive silicates) to decrease collagen degradation [208]. These silicate adhesives showed biological activity when encountered with biological fluid and will produce ionic dissolution products; therefore, they act directly at the level of the hybrid layer. However, this spectacle needs to be further discovered, because in a study, researchers found a decrease in mechanical properties of adhesives and an increase in the bioactivity. Because of the decrease in mechanical properties, bond durability will be compromised [209]. Sauro et al. [202] induced biomimetic remineralization by incorporating phosphoproteins in acid etch dentine and reported that the ion-releasing resin adhesive resulted in biomimetic remineralization due to the release of calcium silicate from adhesive resins [202].

### 6.5. Fluoride

Tooth remineralization also occurs because of an important element known as fluorine [210]. Thus, an adhesive containing fluoride is superior for the hybrid layer strengthening, prevention of secondary caries, and preservation of degradation of the resin dentine bond. Nonetheless, the use of fluoride as a biomimetic analogue is questionable, as biomimetic remineralization requires the seeding of apatite crystallites to remineralize the collagen matrix [60]. Authors in another study incorporated fluoride in resin adhesives and emphasized the significance of fluoride-containing adhesives. They found that, after water storage, the bond strength to dentine was significantly increased [211]. However, there is a dearth of literature in this context, and further investigation will be needed [60,161,209]. Overall, regarding adhesive dentistry, all biomimetic remineralization/biomimetic "smart" materials are new and require further investigations.

## 7. Conclusions

Long-term degradation of dentine-bonded interfaces due to the aging of these adhesives is a major drawback. To enhance the adhesion of resin dentine durability, experimental strategies have been developed with varying success rates. This includes improving the dentine resistance of the collagen matrix to the degradation by enzymes and by preventing collagenolytic activity intrinsically. Using nanotechnologies and other innovative techniques, improvements in the conventional materials and a translation into contemporary materials are needed. These improvements could include the inhibition of MMPs and cathepsins, strengthening of the collagen scaffold, antimicrobial properties, collagen strengthening, and regenerative processes of dental hard tissue. In the next coming years, advancements in technology used to resist collagenolytic hydrolysis and form stable resin–dentine bonds will be available.

## 8. Limitations

After the minimal cavity preparation, when a thicker layer of dentine is affected by the carious lesion, non-homogenous water saturated matrices of resin infiltration occur. The regions where water is in excess and resin is scant will damage over the period of 1–2 years. Moreover, they will undergo fatigue failure due to extreme cyclic strain. To prevent such degradation, demineralized collagen fibers should be remineralized by mineral deposition both intra- and extrafibrillarly. Currently, the dental industry and researchers are more focused on the formulation of the passive adhesive system rather than the development of materials with more clinical longevity that can easily be used in dental practice. These new passive light-curable resin-based materials might cause a toxic reaction by the release of the components.

**Author Contributions:** Conceptualization, F.A., M.A.F. and M.S.Z.; methodology, F.A., M.A.F., M.S.Z. and Z.K.; software and data extraction validation, F.A., M.A.F. and M.S.Z.; manuscript writing, F.A.,

M.A.F., M.S.Z., Z.K. and N.K.; revision and editing, N.K. and P.J.P. All authors have read and agreed to the published version of the manuscript.

**Funding:** This research received no external funding.

**Institutional Review Board Statement:** Not applicable.

**Informed Consent Statement:** Not applicable.

**Data Availability Statement:** Not applicable.

**Conflicts of Interest:** The authors declare no conflict of interest.

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
