# Peer review of "Degradation and Stabilization of Resin-Dentine Interfaces in Polymeric Dental Adhesives: An Updated Review"

_coatings, doi:10.3390/coatings12081094_

Round 1
Reviewer 1 Report
The main objective of the work is giving a review regarding degradation and stabilization of resin-dentine interfaces in polymeric dental adhesives.
In the manuscript the authors offer an overview of the composition and classification of dental adhesives, as well as mechanism of degradation and possible solutions to reduce this unwanted process.
In my opinion the review is well designed and exhaustive.
Just a couple of comments:
a) in the manuscript there is no Figure 2, just Fig.1 and Fig. 3. This misprint must be corrected.
b) the quality of the images presented in the manuscript is low. Please improve the quality and the resolution.
c) in lines 208-212 there are misprints regarding zinc ion molecular formula. Please correct them.
Author Response
REVIEWER 1:
The main objective of the work is giving a review regarding degradation and stabilization of resin-dentine interfaces in polymeric dental adhesives.
In the manuscript, the authors offer an overview of the composition and classification of dental adhesives, as well as the mechanism of degradation and possible solutions to reduce this unwanted process.
In my opinion, the review is well designed and exhaustive.
Author Response: The authors would like to thank the reviewer for their thoughtful and meaningful suggestions that surely helped us to further improve this manuscript. Based on the reviewers’ suggestions, we have revised the manuscript and fully responded to all the Referee’s comments. We have highlighted the changes to the manuscript and are easy to track. Please find a point-by-point response to the reviewer’s comments:
Comment:
In the manuscript there is no Figure 2, just Fig.1 and Fig. 3. This misprint must be corrected.
Authors’ Response: The numbering for all the figures has been corrected.
Comment:
The quality of the images presented in the manuscript is low. Please improve the quality and the resolution
Authors’ Response: All the images are replaced with better-resolution images.
Comment: In lines 208-212 there are misprints regarding the zinc ion molecular formula. Please correct them.
Authors’ Response: Correction has been done (Page 7. Line number 222-226).
The authors would like to thank the reviewer again for the constructive feedback on this article that surely helped to improve the contents and quality. We believe that the quality of the revised manuscript has been improved remarkably and it will be acceptable for publication.
Yours Sincerely,
Reviewer 2 Report
The review is to summarize several studies and the factors important for the degradation of adhesive resins and the strategies to eliminate these factors. The topic is very interesting and has clinical significance. Also, the manuscript is well-written. The manuscript can be considered for publication. I only have a few comments:
Please provide the limitation and outlook in this field.
Please provide the high-resolution of images.
Author Response
REVIEWER 2:
Comment:
The review is to summarize several studies and the factors important for the degradation of adhesive resins and the strategies to eliminate these factors. The topic is very interesting and has clinical significance. Also, the manuscript is well-written. The manuscript can be considered for publication. I only have a few comments:
Authors’ Response: The authors would like to thank the reviewer for their thoughtful and meaningful suggestions that surely helped us to further improve this manuscript. Based on the reviewers’ suggestions, we have revised the manuscript and fully responded to all the Referee’s comments. We have highlighted the changes to the manuscript and are easy to track. Please find a point-by-point response to the reviewer’s comments:
Comment:
Please provide the high-resolution of images.
Authors’ Response: The images is replaced with better resolution image. Page number: 15.
Comment:
Please provide the limitation and outlook in this field.
Authors’ Response: Based on the suggestions, the limitation and outlook in this field are added (line number 695-704).
The authors would like to thank the reviewer again for the constructive feedback on this article that surely helped to improve the contents and quality. We believe that the quality of the revised manuscript has been improved remarkably and it will be acceptable for publication.
Yours Sincerely,
Reviewer 3 Report
This is a review focused on the durability of dentin-adhesive interfaces and various material technologies to make more durable this interface. This is an important area of research. I have a few comments to improve this review but am happy to see it published.
Page 2, line 72-72 – Rephrase sentence
Table 1 – Define all abbreviations
The authors should note that the “generation” idea is based on marketing and is a very, very general way, imprecise way, to categorize adhesives and should thus be used with caution.
It is a little strange that Figure 1 is an MMP? Why not an adhesive, bonded interface or something more informative? Could the authors comment on what clinical evidence there is for the role that MMPs play in degradation?
Biomimetic remin section should discuss the importance or intra vs inter fibrillar mineralization of collagen.
Figure 3 has green triangles that should be removed.
Figure 2 is missing? Please, fix this.
Author Response
REVIEWER 3
This is a review focused on the durability of dentin-adhesive interfaces and various material technologies to make more durable this interface. This is an important area of research. I have a few comments to improve this review but am happy to see it published.
Authors’ Response: Authors would like to thank the respected reviewer for reviewing and providing insightful and constructive feedback on the manuscript that certainly helped us to improve the content. The authors worked thoroughly and revised the manuscript based on the provided suggestions. Please find a point-by-point response to the comments below:
Comment:
Page 2, lines 72-72 – Rephrase sentence
Authors’ Response: The mentioned statement has been rephrased (Page 2, lines 74-76).
Comment:
Table 1 – Define all abbreviations
Authors’ Response: All the abbreviations in Table 1 are defined now (Table 1) (Page 3 line number 121-122).
Comment:
The authors should note that the “generation” idea is based on marketing and is a very, very general way, imprecise way, to categorize adhesives and should thus be used with caution.
Authors’ Response: The authors totally agree with the reviewer’s comment, therefore in addition to this generation-based classification, all other basic classifications of adhesives have been mentioned in the table. Moreover, sub-classification according to the number of steps have been added to the table simultaneously. But the generation-based classification is so much used among the dental fraternity that we cannot ignore it.
Comment:
It is a little strange that Figure 1 is an MMP? Why not an adhesive, bonded interface or something more informative? Could the authors comment on what clinical evidence there is for the role that MMPs play in degradation?
Authors’ Response: The basic knowledge about adhesives their classification composition and historical background have been given in the initial part of the manuscript. However, the enzyme, matrix metalloproteinases (MMPs) has a crucial role in the degradation of type I collagen, the organic component of the hybrid layer. It has been found that the hydrophilic and acidic characteristics of current dentin adhesives have made hybrid layers highly prone to water sorption. This, in turn, causes polymer degradation and results in decreased resin–dentin bond strength over time. These unstable polymers inside the hybrid layer may result in denuded collagen fibers, which become vulnerable to mechanical and hydrolytic fatigue, as well as degradation by host-derived proteases with collagenolytic activity.
Comment:
Biomimetic remin section should discuss the importance or intra vs inter fibrillar mineralization of collagen.
Authors’ Response: The suggested information has been added (page 28; lines 595-596 and 607-610).
Comment:
Figure 3 has green triangles that should be removed.
Authors’ Response: Green triangles have been removed from Figure 3.
Comment:
Figure 2 is missing? Please, fix this.
Authors’ Response: We fix the image number.
The authors would like to thank the reviewer again for the constructive feedback on this article that surely helped to improve the contents and quality. We believe that the quality of the revised manuscript has been improved remarkably and it will be acceptable for publication.
Yours Sincerely,
This manuscript is a resubmission of an earlier submission. The following is a list of the peer review reports and author responses from that submission.